# Prognostic Value of Initial Diagnostic Imaging Findings for Patient Outcomes in Adult Patients with Traumatic Brain Injury: A Systematic Review and Meta-Analysis

Hang Yu, Sudharsana Rao Ande , Divjeet Batoo, Janice Linton and Jai Shankar *

Department of Radiology, University of Manitoba, GA216-820 Sherbrook Street, Winnipeg, MB R3A 1R9, Canada
* Correspondence: jshankar@hsc.mb.ca

**Abstract:** Introduction: Termed the "silent epidemic," traumatic brain injury (TBI) is one of the greatest global contributors not only to post-traumatic death but also to post-traumatic long-term disability. This systematic review and meta-analysis aims to specifically evaluate the prognostic value of features on initial imaging completed within 24 h of arrival in adult patients with TBI. Method: The authors followed the PRISMA 2020 checklist for systematic review and meta-analysis design and reporting. Comprehensive searches of the Medline and Embase databases were carried out. Two independent readers extracted the following demographic, clinical and imaging information using a predetermined data abstraction form. Statistics were performed using Revman 5.4.1 and R version 4.2.0. For pooled data in meta-analysis, forest plots for sensitivity and specificity were created to calculate the diagnostic odds ratio (DOR). Summary receiver operating characteristic (SROC) curves were generated using a bivariate model, and diagnostic accuracy was determined using pooled sensitivity and specificity as well as the area under the receiver operator characteristic curve (AUC). Results: There were 10,733 patients over the 19 studies. Overall, most of the studies included had high levels of bias in multiple, particularly when it came to selection bias in patient sampling, bias in controlling for confounders, and reporting bias, such as in reporting missing data. Only subdural hematoma (SDH) and mortality in all TBI patients had both an AUC with 95% CI not crossing 0.5 and a DOR with 95% CI not crossing 1, at 0.593 (95% CI: 0.556–0.725) and 2.755 (95% CI: 1.474–5.148), respectively. Conclusion: In meta-analysis, only SDH with mortality in all TBI patients had a moderate but significant association. Given the small number of studies, additional research focused on initial imaging, particularly for imaging modalities other than NECT, is required in order to confirm the findings of our meta-analysis and to further evaluate the association of imaging findings and outcome.

**Keywords:** traumatic brain injury; diagnostic imaging; systematic review



## 1. Introduction

Termed the "silent epidemic," traumatic brain injury (TBI) is one of the greatest global contributors not only to post-traumatic death but also to post-traumatic long-term disability in multiple domains ranging from neurological and physical to behavioral and psychosocial [1–3]. On a per capita basis, the United States and Canada together demonstrate the highest incidence of TBI in the world at 1299 cases per 100,000 people [1]. It is estimated that 1.1% of the American population, or roughly 3.1 million people, are living with long-term disability post-TBI associated with a lifetime economic cost of $76.5 billion in 2010 US dollars [3,4]. Despite this, the highest healthcare and socioeconomic burden is seen in resource-poor and lower-income countries [1].

Since its adoption around 50 years ago, non-enhanced computed tomography (NECT) of the head has become an indispensable and widely available imaging tool that can accurately and promptly diagnose intracranial injuries post-TBI including intracranial

hemorrhage (ICH), mass effect and herniation, midline shift, and cranial fractures [5]. To aid in predicting outcomes in TBI patients based on NECT findings, classification systems such as the Marshall score (in 1991) and the Rotterdam CT score in (2005) have been developed [6,7]. Multivariate models combining both clinical and imaging findings such as the International Mission on Prognosis and Analysis of Clinical Trials in TBI (IMPACT) also attempt to prognosticate, but they are not widely used clinically and are designed for guiding clinical trials [8]. In addition to NECT, other CT techniques such as CT perfusion (CTP) [9], and additional imaging modalities such as positron emission tomography (PET) [10], single-photon emission computed tomography (SPECT) [11], transcranial doppler (TCD) ultrasound (US) [12], and magnetic resonance imaging (MRI) [13], have also been used to evaluate the extent of intracranial injuries post-TBI and to aid in prognostication.

This systematic review and meta-analysis aims to specifically evaluate the prognostic value of features on initial imaging completed within 24 h of arrival in adult patients with TBI.

## 2. Methods

The authors followed the PRISMA 2020 checklist for systematic review and meta-analysis design and reporting [14]. Comprehensive searches of the Medline and Embase databases were carried out by an experienced librarian (JL) using the OVID platform to identify relevant studies for inclusion. A combination of keyword and MeSH or EMTREE subject headings was used.

The search was limited to articles published in peer-reviewed journals in English from inception to September 2020. All imaging modalities were eligible for inclusion. Aside from reviews, editorials and case reports, all study types and designs were included. Studies were excluded if (1) there were no patients with TBI OR (2) initial imaging was not performed within 24 h of arrival to emergency department (ED) OR (3) there was no follow-up or patient outcome measures reported OR (4) the mechanism of insult was non-traumatic OR (5) patients less than 18 years of age were included OR (6) the study sample size was fewer than five individuals.

Three independent screeners extracted the following information using a predetermined data abstraction form: study characteristics, patient demographics, information regarding patient initial clinical presentation when available (e.g., Glasgow coma score (GCS), pupil reactivity and evenness, vital signs), imaging modality characteristics, associated imaging findings (e.g., ICH, midline shift), and follow-up and associated patient outcomes related to study imaging findings.

### 2.1. Statistics

Statistics were performed using Revman 5.4.1 (Cochrane, London, UK) and R version 4.2.0 (R Foundation for Statistical Computing, Vienna, Austria) [15,16]. Each imaging finding and associated outcome measure had $2 \times 2$ tables created to calculate the sensitivity, specificity, positive predictive value (PPV), and negative predictive value (NPV). For pooled data in meta-analysis, forest plots for sensitivity and specificity were created to calculate the diagnostic odds ratio (DOR). Summary receiver operating characteristic (SROC) curves were generated using a bivariate model, and diagnostic accuracy was determined using pooled sensitivity and specificity as well as the area under the receiver operator characteristic curve (AUC). The R program "mada" was used to generate the SROC curves. AUC and 95% confidence intervals (CI) were calculated using a parametric bootstraps method [17]. Study heterogeneities were measured by Higgins' $I^2$.

### 2.2. Assessing Bias

Bias was assessed using the domains of the ROBINS-I tool (Cochrane, London, UK) [18]. Two authors independently assessed each domain for high, unclear, and low risk of bias in each included study. Disagreements were resolved with consensus. Inclusion

of only studies with full publication in peer-reviewed journals was done to maintain the highest possible quality of evidence.

## 3. Results

The study selection process is shown in Figure 1. The initial search yielded 4586 results, and 4027 unique articles after duplicates were removed. Of the original 109 articles, only 19 studies were eligible for inclusion within our systematic review [19–37]. A summary of the articles included can be found in Table 1. In total, there were 10,733 patients over the 19 studies. The risk of bias analysis is shown in Figure 2. Overall, most of the included studies had high levels of bias in multiple, particularly when it came to selection bias in patient sampling, bias in controlling for confounders, and reporting bias, such as in reporting missing data.

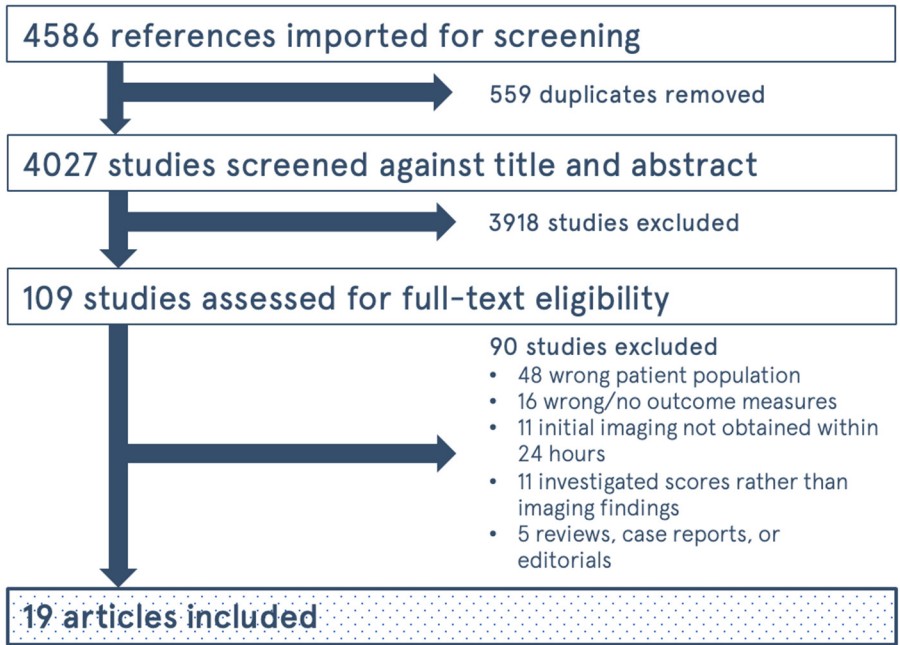

**Figure 1.** Flow Diagram for study selection process.

**Table 1.** Study characteristics.

| First Author and Year | Study Design | # of Patients | Imaging Modality | Patient Outcome Measure(s) |
|---|---|---|---|---|
| Bindu et al. 2017 [19] | Prospective cohort | 78 | CTP | GOS at 3-month follow-up. <br> ○ GOS 1–3: unfavourable outcome. <br> ○ GOS 4–5: favourable outcome. |
| Compagnone et al. 2009 [20] | Prospective cohort | 315 | NECT | GOS at 6-month follow-up. <br> ○ GOS 1–3: unfavourable outcome. <br> ○ GOS 4–5: favourable outcome. |
| Henninger et al. 2018 [21] | Prospective cohort | 361 | NECT | ICU length of stay and complications. <br> Survival at discharge, and 3-month and 12-month follow-up. <br> GOS at 3-month and 12-month follow-up. <br> ○ GOS 1–3: unfavourable outcome. <br> ○ GOS 4–5: favourable outcome. |

**Table 1.** *Cont.*

| First Author and Year | Study Design | # of Patients | Imaging Modality | Patient Outcome Measure(s) |
|---|---|---|---|---|
| Khalilabadi et al. 2016 [22] | Prospective cohort | 100 | NECT | GOS (unspecified follow-up interval).<br>○ GOS 1–2: unfavourable outcome.<br>○ GOS 3–5: favourable outcome. |
| Kotwica and Jakubowski 1995 [23] | Retrospective cohort | 111 | NECT | GOS 60 days post-admission. |
| Kreitzer et al. 2017 [24] | Retrospective cohort | 240 | NECT | Mortality within 2 weeks. Social security death index for no follow-up after 14 days post-injury. Neurosurgical intervention (procedure performed by neurosurgery for head injury, including external ventricular drain (EVD) placement or intracranial pressure (ICP) monitor placement) within 2 weeks. Length of stay >48 h. |
| Legrand et al. 2013 [25] | Prospective cohort | 77 | NECT | Mortality during ICU admission.<br>GOS at 6-month follow-up.<br>○ GOS 1–3: unfavourable outcome.<br>○ GOS 4–5: favourable outcome. |
| Letourneau-Guillon et al. 2013 [26] | Retrospective cohort | 66 | CTA with delayed phase | Hematoma expansion (>12 mL and 33%) Need for hematoma surgical evacuation and in-hospital mortality.<br>○ Poor outcome defined as at least one of the above is present. |
| Moreno et al. 2000 [27] | Prospective cohort | 125 | TCD and NECT | GOS at 6-month follow-up.<br>○ GOS 1–3: unfavourable outcome.<br>○ GOS 4–5: favourable outcome. |
| Naraghi et al. 2015 [28] | Retrospective cohort | 600 (132) | NECT (with CTA) | Primary: Changes to management or additional medical therapy by CTA findings.<br>Secondary: Admission to ICU, ICU length-of-stay, hospital length-of-stay, discharge disposition, in-hospital mortality. |
| Quigley et al. 2013 [29] | Retrospective cohort | 478 | NECT | Length of ICU and hospital stay, progression of hemorrhage and other injuries, any need for further intervention or clinical deterioration, in-hospital mortality, discharge home status. |
| Shankar et al. 2020 [30] | Pilot prospective cohort | 19 | NECT with CTA and CTP | In-hospital mortality ≤48 h of admission. |
| Splavski et al. 2006 [31] | Prospective cohort | 30 | TCD | GOS at 6-month follow-up.<br>○ GOS 1–3: unfavourable outcome.<br>○ GOS 4–5: favourable outcome. |
| Tasaki et al. 2009 [32] | Prospective cohort | 111 | NECT | GOS at 6-month follow-up.<br>○ GOS 1–3: unfavourable outcome.<br>○ GOS 4–5: favourable outcome. |

| First Author and Year | Study Design | # of Patients | Imaging Modality | Patient Outcome Measure(s) |
|---|---|---|---|---|
| Tucker et al. 2017 [33] | Prospective cohort | 7277 | NECT | In-hospital mortality. |
| Waqas et al. 2015 [34] | Retrospective cohort | 117 | NECT | ICU length of stay, total length of stay, survival at discharge.<br>GOS at latest follow-up.<br>○ GOS 1–3: unfavourable outcome.<br>○ GOS 4–5: favourable outcome. |
| Wintermark et al. 2004 [35] | Prospective cohort | 130 | NECT with CTP | GOS at 3-month follow-up.<br>○ GOS 1 split to 1a (death due to primary lesion).<br>and 1b (death due to late complication). |
| Wong et al. 2009 [36] | Retrospective cohort | 464 | NECT | Length of ICU stay, and total hospital stay.<br>In-hospital mortality and 1-year mortality.<br>GOS at 1-year follow-up.<br>○ GOS 1–3: unfavourable outcome.<br>○ GOS 4–5: favourable outcome. |
| Yamamura et al. 2016 [37] | Prospective cohort | 34 | NECT | In-hospital mortality.<br>GOS during admission.<br>○ GOS 2<br>○ GOS 3–5 |

CTA: CT angiography; CTP: CT perfusion; GOS: Glasgow outcome scale; NECT: Non-enhanced CT; TCD: transcranial doppler ultrasound.

Imaging modalities that were investigated included NECT, CTP, CTA, and TCD. Patient outcome measures and follow-up intervals varied widely, but most of the included studies used the Glasgow Outcome Scale (GOS) as their primary patient outcome measure in follow-up, with a GOS 1–3 commonly interpreted as an unfavorable outcome and a GOS 4–5 interpreted as a favorable outcome. Only data involving the association between imaging features and patient outcome measures were recorded. In certain studies, the authors report imaging features as demographic information but ultimately did not provide data or analysis on their association with patient outcome measures. These were not included in analysis. Some studies, in addition to researching the association between imaging findings and patient outcome measures, also investigated imaging finding associations with non-outcome-related findings including biochemical and laboratory data, such as lactate levels, and relationships with other imaging findings. These were also not included. Isolated relationships between imaging and patient clinical features, such as CTP findings with intracranial hypertension, were not recorded unless an associated patient outcome measure was also analyzed.

The association between imaging findings and select patient outcome measures is summarized in Table 2. In general, the presence of subarachnoid hemorrhage (SAH), midline shift, cerebral edema, and basal cistern effacement are associated with poorer outcomes in TBI. Other types of ICH were not associated with worse outcomes, especially after consideration of confounders in multivariate analysis. Kotwica and Jakubowski 1995 found that the presence of subdural hematoma (SDH) was associated with worse outcomes and epidural hematoma (EDH) was associated with better outcomes in patients presenting with GCS 3 [23]. While the presence of diffuse axonal injury (DAI) on CT was associated with worse outcomes in comparison to the absence of DAI on CT, predominantly hemorrhagic DAI on CT was associated with better outcomes compared to predominantly non-hemorrhagic DAI [21].

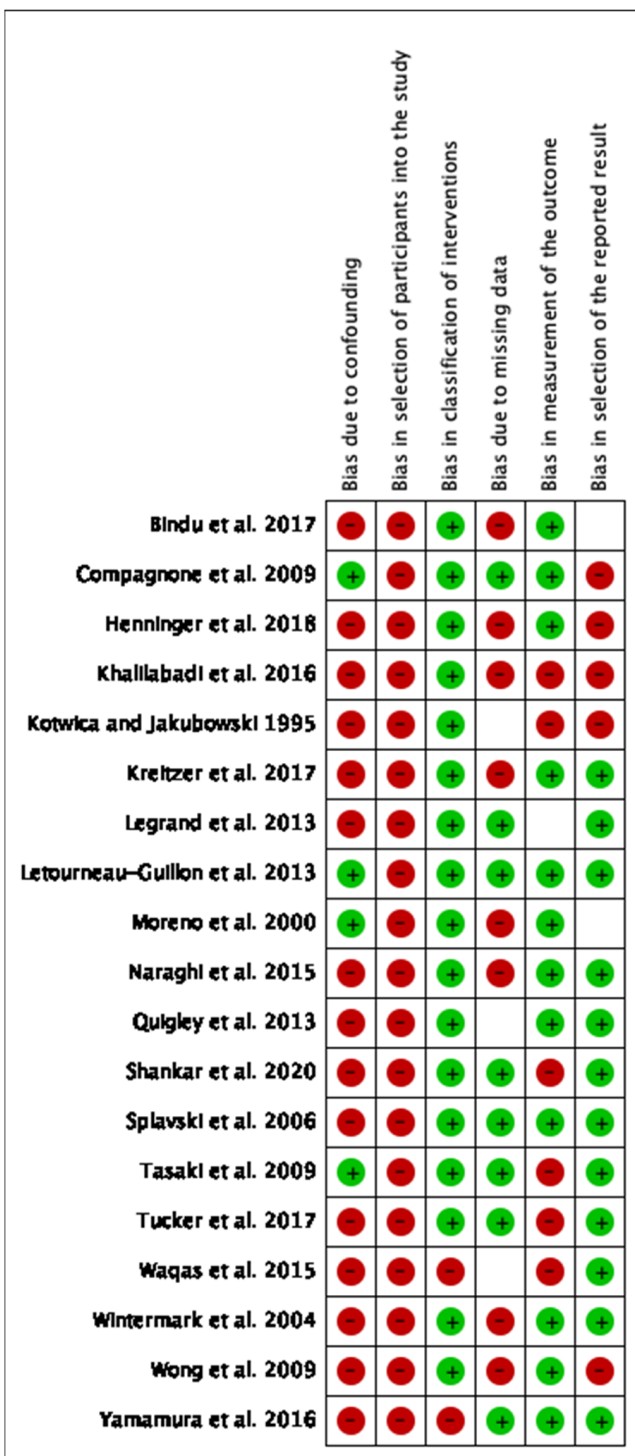

**Figure 2.** Risk of bias analysis for the included studies [19–37]. Red indicates high risk of bias, blank indicates unclear risk of bias, and green indicates low risk of bias.

**Table 2.** Imaging findings and their association with patient outcomes.

| Imaging Finding | Relationship with Patient Outcome |
|---|---|
| Subarachnoid hemorrhage (SAH) | Association with worse outcomes<br>○ Compagnone 2009 [20]: OR: 1.80 (95% CI: 0.97–3.35; *p* = 0.046) for GOS 1–3 at 6-month FU.<br>○ Kreitzer 2017 [24]: SAH predictor of adverse outcomes preventing discharge from EDs in TBI patients presenting with GCS 15.<br>○ Legrand 2013 [25]: SAH associated with in-hospital mortality during ICU admission.<br>○ Moreno 2000 [27]: Absent SAH OR: 0.45 (95% CI: 0.21–0.94; *p* = 0.03) for GOS 1–3 at 6-month FU.<br>○ Tasaki 2009 [32]: Extensive SAH (defined as presence of both convexal and basal cistern SAH) significantly associated with GOS 1–3 in 6-month FU (*p* < 0.001) in univariate analysis. Multivariate logistic regression OR 31.1 (95% CI: 2.36–410; *p* = 0.01).<br>No significant association with outcome<br>○ Letourneau-Guillon 2013 [26]: *p* = 0.122 for hematoma expansion (>12 mL and 33%) or need for hematoma surgical evacuation or in-hospital mortality.<br>○ Quigley 2013 [29]: Isolated SAH (i.e., no other pathological intracranial finding) on NECT considered benign. |
| Subdural hematoma (SDH) | Association with worse outcomes<br>○ Kotwica 1995 [23]: SDH associated with mortality in GCS 3 patients.<br>No significant association with outcome<br>○ Letourneau-Guillon 2013 [26]: *p* = 0.233 for hematoma expansion (>12 mL and 33%) or need for hematoma surgical evacuation or in-hospital mortality.<br>○ Wong 2009 [36]: In those with intracerebral hematoma, an associated SDH has OR 2.631 (95% CI: 0.841–9.373; *p* = 0.096) for in-hospital mortality.<br>    ● No significance for 1-year GOS 4–5.<br>    ● No significance for 1-year mortality after multivariate analysis OR 2.93 (95% CI: 0.952–9.007; *p* = 0.061). |
| Epidural hematoma (EDH) | Association with better outcomes<br>○ Kotwica 1995 [23]: EDH independent predictor of survival over 60 days in GCS 3.<br>No significant association with outcome<br>○ Letourneau-Guillon 2013 [26]: *p* = 0.483 for hematoma expansion (>12 mL and 33%) or need for hematoma surgical evacuation or in-hospital mortality. |
| Intraparenchymal hematoma (IPH)/Contusion | No significant association with outcome<br>○ Letourneau-Guillon 2013 [26]: *p* = 0.367 for hematoma expansion (>12 mL and 33%) or need for hematoma surgical evacuation or in-hospital mortality. |
| Intraventricular hemorrhage (IVH) | No significant association with outcome<br>○ Compagnone 2009: OR: 1.17 (95% CI: 0.15–7.07; *p* = 1.00) for GOS 1–3 at 6-month FU for GCS 9–13.<br>○ Letourneau-Guillon 2013 [26]: *p* = 0.056 for hematoma expansion (>12 mL and 33%) or need for hematoma surgical evacuation or in-hospital mortality. |
| Hematoma size | Association with worse outcomes<br>○ Letourneau-Guillon 2013 [26]: 4.44 mL (IQR: 1.65–12.8) vs. 32.9 mL (IQR: 8.46–82.9) *p* < 0.01 for good outcome vs. poor outcome (hematoma expansion (>12 mL and 33%) or need for hematoma surgical evacuation or in-hospital mortality) respectively.<br>No significant association with outcome<br>○ Wong 2009 [36]: Hematoma volume >50 mL and >30 mL no significant association with in-hospital mortality, 1-year mortality, or 1-year GOS 4–5 after multivariate analysis. |

**Table 2.** *Cont.*

| Imaging Finding | Relationship with Patient Outcome |
|---|---|
| Hematoma location | Wong 2009 [36]:<br>○ Temporal location of intracerebral hematoma OR 2.631 (95% CI: 0.841–9.373; $p = 0.096$) in-hospital mortality.<br>○ Bilateral intracerebral hematoma OR 0.374 (95% CI: 0.120–1.172; $p = 0.092$) for 1-year GOS 4–5.<br>○ No significance association of frontal location and side of intracerebral hematoma in univariate analysis with in-hospital mortality. No significance association of bilateral intracerebral hematoma in multivariate analysis with in-hospital mortality. |
| Diffuse axonal injury (DAI) | Henninger 2018 [21]:<br>○ Predominant hemorrhagic DAI (where DAI is an isolated/the predominant intracranial abnormality i.e., no mass lesion, other large hemorrhage) not independent predictor of poor outcome.<br>○ Predominant hemorrhagic DAI associated with better outcomes vs. associated hemorrhage DAI (i.e., DAI is found with other significant intracranial findings).<br>    • Discharge survival OR 24.7 (95%: CI 3.2–192.6; $p = 0.002$).<br>    • 1-year GOS 4–5 adjusted OR 4.7 (95% CI:1.5–15.2; $p = 0.01$).<br>    • Significantly less common associated brain herniation ($p < 0.001$).<br>○ Predominant hemorrhagic DAI associated with significantly longer ICU stay (median 16 vs. 7 days, $p < 0.001$) and increased ICU complications (e.g., sepsis, pneumonia, ARDS) compared to CT-DAI negative patients. |
| Midline shift | Association with worse outcomes<br>○ Letourneau-Guillon 2013 [26]: Midline shift ≥5 mm significantly associated with poor outcome $p < 0.01$ (Hematoma expansion (>12 mL and 33%) or need for hematoma surgical evacuation or in-hospital mortality).<br>    • OR 13.77 (95% CI: 1.54–123.49; $p < 0.020$) for in-hospital mortality.<br>○ Tasaki 2009 [32]: Midline shift (any) significantly associated with GOS 1–3 in 6-month FU $p < 0.001$ in univariate analysis.<br>No significant association with outcome<br>○ Waqas 2015 [34]: $p = 0.342$ for midline shift ≥5 mm and mortality in TBI patients needing decompressive craniotomy. |
| Cerebral edema | Association with worse outcomes<br>○ Tucker 2017 [33]: OR 8.02 (95% CI: 4.60–14.00; $p < 0.0001$) for in-hospital mortality in all TBI patients after logistic regression. OR 4.88 (95% CI: 2.03–11.75; $p = 0.0004$) in mild (GCS 13–15) TBI patients.<br>    • Uncorrected OR 18.1 all TBI patients. 11.4 in mild (GCS 13–15), 10.2 In moderate (GCS 9–12), 2.3 severe (GCS ≤8).<br>    • Uncorrected OR 76.6 for patients with no ICH.<br>○ Wintermark 2004 [35]: Independent prognostic factor for GOS 1–3 at 3-month FU $p = 0.041$.<br>No significant association with outcome<br>○ Waqas 2015 [34]: $p = 0.624$ for mortality in TBI patients needing decompressive craniotomy. |
| Basal cistern effacement/compression | Association with worse outcomes<br>○ Tasaki 2009 [32]: Absent suprasellar cisterns, ambient cisterns, and quadrigeminal cisterns significantly associated with GOS 1–3 in 6-month FU $p < 0.001$ in univariate analysis. |
| Brain herniation | Association with worse outcomes<br>○ Wintermark 2004 [35]: Independent prognostic factor for GOS 1–3 at 3-month FU $p = 0.013$. |

**Table 2.** *Cont.*

| Imaging Finding | Relationship with Patient Outcome |
|---|---|
| Skull fracture | No significant association with outcome<br>○ Compagnone 2009: OR: 1.15 (95% CI: 0.57–2.29) $p = 0.68$ for GOS 1–3 at 6-month FU. |
| Angiography findings | Letourneau-Guillon 2013 [26]:<br>○ Contrast extravasation significantly associated with poor outcome (hematoma expansion >12 mL and >33% or hematoma evacuation or in-hospital mortality) $p < 0.01$.<br>○ Contrast extravasation associated with poorer outcomes regardless of hematoma type (SAH, SDH, EDH, IVH, or IPH).<br>○ Contrast extravasation significantly associated with mortality OR 8.00 (95% CI: 2.47–25.9; $p = 0.001$) on univariate analysis.<br>   ● On multivariate logistic regression, OR 4.48 (95% CI: 1.31–15.29; $p = 0.017$)<br>○ Any pattern of contrast extravasation associated with surgical evacuation OR 3.88 (95% 1.64–9.21) $p = 0.0021$).<br>○ Contained or active extravasation on arterial phase imaging significantly associated with need for evacuation. OR 3.75 (95% CI: 1.62–8.72; $p = 0.002$) and OR 4.57 (95% 1.81–11.56; $p = 0.0013$).<br>   ● No extravasation on arterial phase and only seen on delayed phase not associated OR 1.02 (95% CI: 0.31–3.37; $p = 0.972$).<br>Naraghi 2015 [28]:<br>○ 33/132 patients had additional findings on CTA: nonacute vascular malformations, compression of small intracranial vessels from the mass effect of the injury, irregularities in opacification of nonessential vessels that could have been small injuries.<br>   ● No significant alteration in management in hospital or during follow-up with neurology to date of publication.<br>○ CTA findings only altered management of only 1/132 patients, who had a clinically significant arteriovenous malformation after isolated SAH in ambient cistern post fall.<br>○ Recommended against the use of CTA in initial TBI imaging. |
| CT Perfusion findings | Bindu 2017 [19]<br>○ Higher mean whole brain CBF and CBV associated with higher GOS.<br>○ Reduced frontal and occipital lobe CBF and CBV associated with poorer GCS on arrival.<br>○ CBF of frontal area showed better correlation with GOS.<br>○ CBF was the most important predictor among all the perfusion parameters.<br>Shankar 2020 [30]<br>○ Decreased whole brain CBF and CBV had sensitivity of 50%, specificity of 100%, PPV of 100%, NPV 88.23% and AUC 0.75 for in-hospital mortality ≤48 h of admission.<br>○ Decreased brainstem CBF and CBV had sensitivity of 75%, specificity of 100%, PPV of 100%, NPV 93.75% and AUC 0.87 for in-hospital mortality ≤48 h of admission.<br>Wintermark 2004 [35]<br>○ Number of arterial territories (out of 6 corresponding to the 2 ACA, MCA and PCA territories) with decreased CBV independent prognostic factor for GOS 1–3 at 3-month FU $p = 0.008$.<br>○ Decreased CBV and CBF and increased MTT in 1 or more vascular territory associated with GOS 1a and 2. |
| MCA velocity and PI | Moreno 2000 [27]:<br>○ Mean MCA velocity in GOS 4–5 at 6-month FU was 44 cm/s vs 36 cm/s in GOS 1–3 at 6-month FU ($p = 0.003$).<br>○ Higher PI OR 21.42 (95% CI: 3.81–183.08; $p = 0.001$) for GOS 1–3 6-months post injury on multivariate analysis.<br>○ PI >2.3 associated with 100% mortality rate.<br>Splavski 2006 [31]:<br>○ Weak but statistically significant positive correlation of higher MCA and higher GOS (r = 0.136; $p < 0.01$).<br>○ Strong statistically significant negative correlation of higher PI and higher GOS (r = −0.70722; $p < 0.01$). |

**Table 2.** *Cont.*

| Imaging Finding | Relationship with Patient Outcome |
|---|---|
| Optic nerve sheath diameter (ONSD) | Legrand 2013 [25]:<br><br>○ Higher ONSD (6.8 ± 0.1 mm vs. 7.8 ± 0.1 mm) in those who died during ICU admission ($p < 0.001$).<br>○ ROC curve AUC for ONSD: 0.805 (95% CI 0.694–0.883).<br>○ ONSD ≥ 7.3 mm had a sensitivity 86.4% (95% CI: 65.1%–97.1%), specificity of 74.6% (95% CI: 61.0%–85.3%), PPV 57.6% (95% CI: 38.9%–74.8%), NPV of 93.2% (81.3%–98.6%), a LR+ of 3.4 (95% CI: 2.7–4.3) and a LR- of 0.2 (95% CI: 0.1–0.6).<br>○ Higher ONSD associated with lower GOS 6-month post injury ($p = 0.03$).<br><br>Waqas 2015 [34]:<br><br>○ AUC for bilateral mean ONSD for mortality 0.49 (95% CI: 0.36—0.62) in TBI patients needing decompressive craniotomy.<br>○ Associated with increased ICP but does not predict mortality or unfavourable outcomes in those requiring decompressive craniotomy. |

ACA: anterior cerebral artery; ARDS: acute respiratory distress syndrome; AUC; area under the curve; CBF: cerebral blood flow; CBV: cerebral blood volume; CTA: CT angiography; CTP: CT perfusion; DAI: diffuse axonal injury; EDH: epidural hematoma; ED: emergency department; FU: follow-up; GCS: Glasgow coma score; GOS: Glasgow outcome scale; ICH: intracranial hemorrhage; ICP: intracranial pressure; IPH: intraparenchymal hematoma; IVH: intraventricular hemorrhage; LR: likelihood ratio; MCA: middle cerebral artery; NECT: Non-enhanced CT; NPV; negative predictive value; ONSD: optic nerve sheath diameter; PCA: posterior cerebral artery; PI: pulsatility index; PPV; positive predictive value; ROC; receiver operating characteristics; SAH: subarachnoid hemorrhage; SDH: subdural hematoma; TBI: traumatic brain injury.

In CTP, decreased cerebral blood flow (CBF) and cerebral blood volume (CBV) prognosticated poorer outcomes and higher mortality [19,30,35]. Increased pulsatility index (PI) and decreased middle cerebral artery (MCA) velocity on TCD were associated with poorer outcomes, as was increased optic nerve sheath diameter (ONSD) [25,27,31]. Increased ONSD was not associated with worse outcomes in a subpopulation consisting of those requiring decompressive craniectomy [34]. Regarding findings on CT angiography (CTA), Naraghi et al. 2015 found that findings on CTA only changed management for one of their 132 patients and concluded that the examination is unnecessary as an initial investigation for TBI [28]. Alternatively, Letourneau-Guillon et al. 2013 found that arterial extravasation predicted the need for surgical evacuation and in-hospital mortality [26].

The data for nine studies [22,23,25,33,34,36] could be pooled for meta-analysis. Figure 3 provides the forest plots for sensitivity and specificity for the meta-analysis data.

The results of Figure 3 were used to generate the SROC curves shown in Figure 4. The summary estimate represents the mean sensitivity and 1-specificity (false positive rate) of the pooled data for each imaging finding with an associated 95% confidence region outlined by the thin black line. Using the SROC curves, the diagnostic odds ratio (DOR) and the area under the receiver operator characteristic curve (AUC) were calculated along with their 95% confidence intervals (CI).

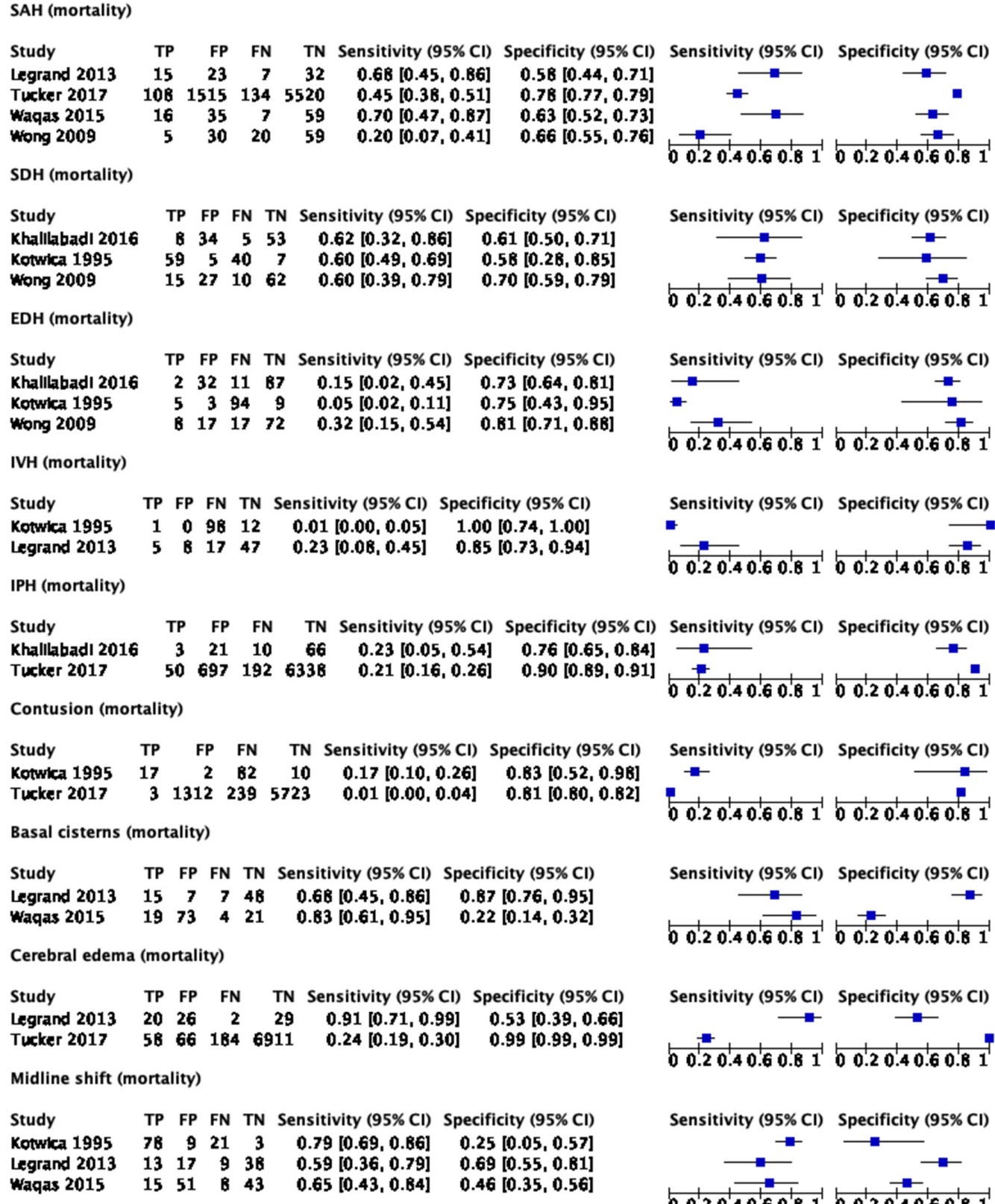

**Figure 3.** Forest plots for imaging findings and mortality in all TBI patients [22,23,25,33,34,36]. TP: true positive; FP: false positive; FN: false negative; TN: true negative.

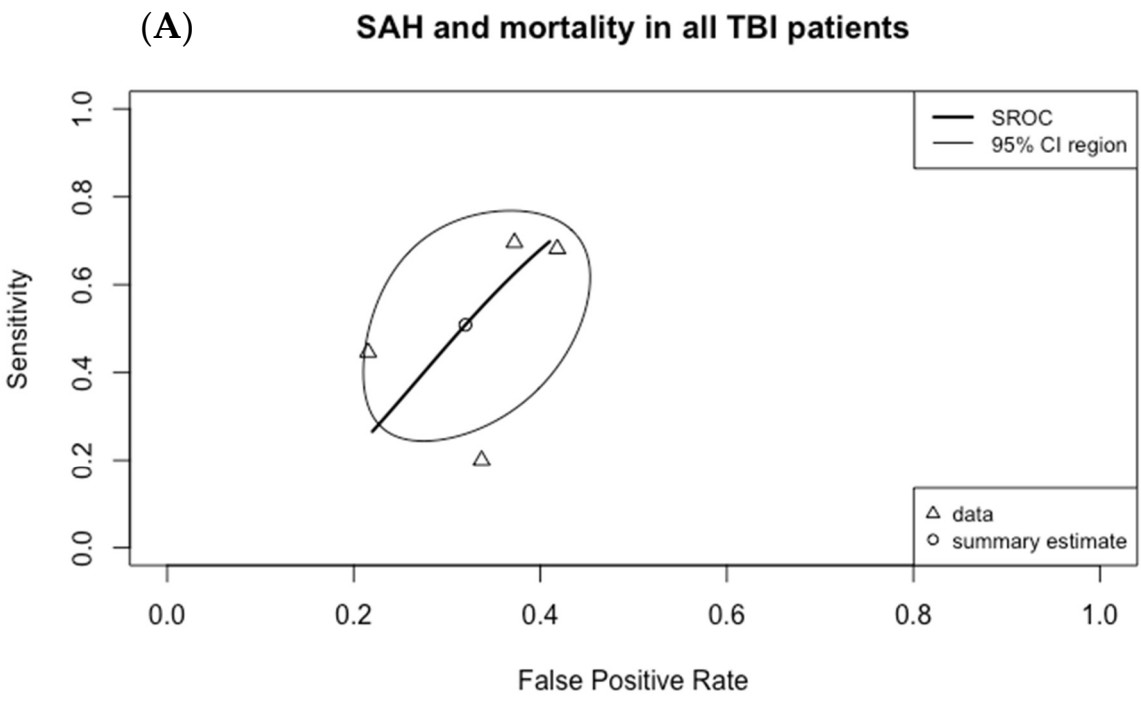

AUC: 0.658 (95%: 0.467–0.749).
DOR: 2.156 (95% CI: 1.024–4.543)
Higgins' $I^2$: 26.263%

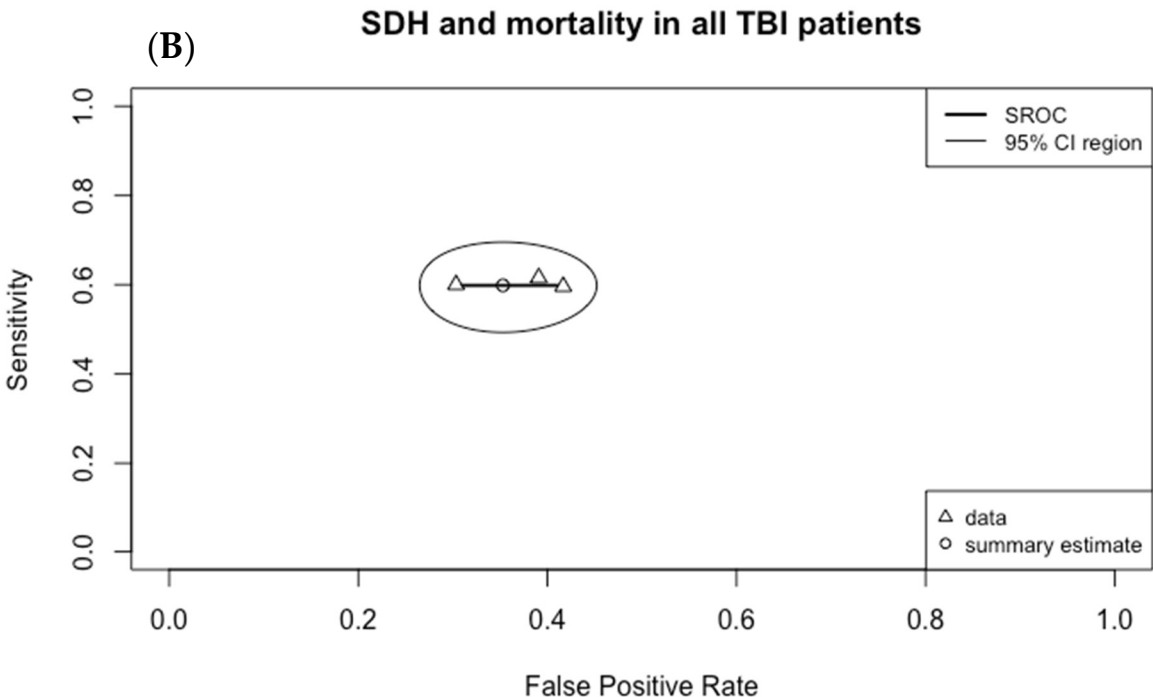

AUC: 0.593 (95% CI: 0.556–0.725).
DOR: 2.755 (95% CI: 1.474–5.148).
Higgins' $I^2$: 0%

**Figure 4.** *Cont.*

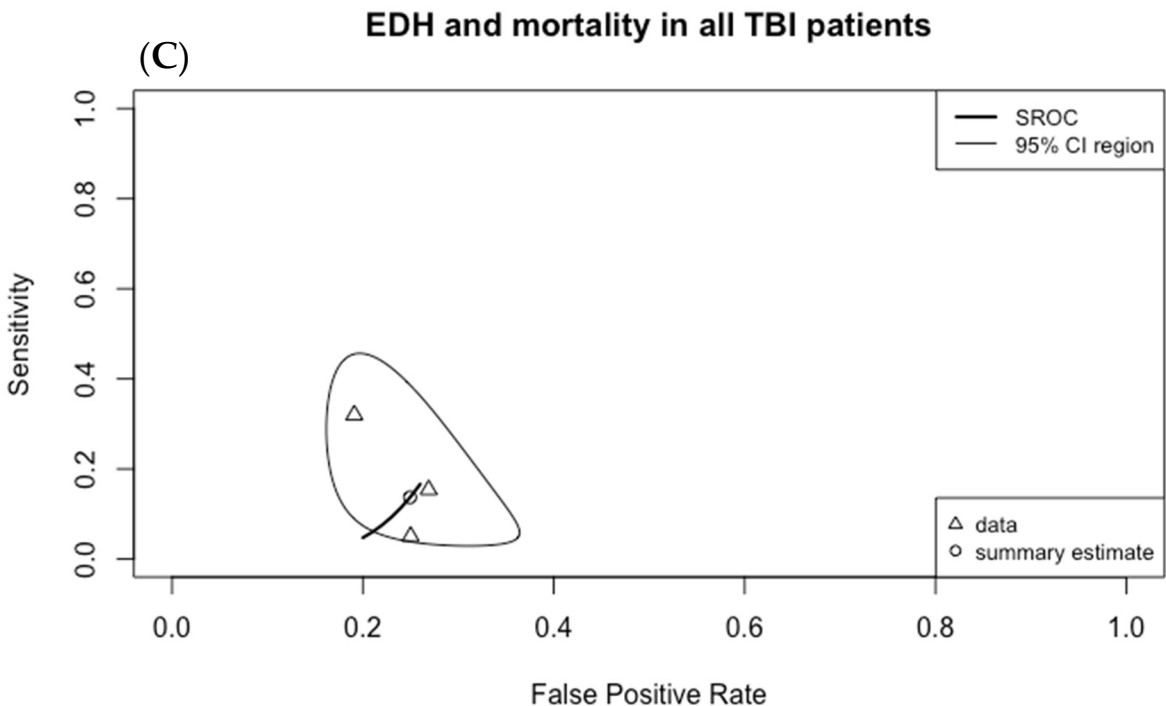

AUC: 0.647 (95% CI: 0.136–0.789)
DOR: 0.596 (95% CI: 0.129–2.765).
Higgins' I2: 0%

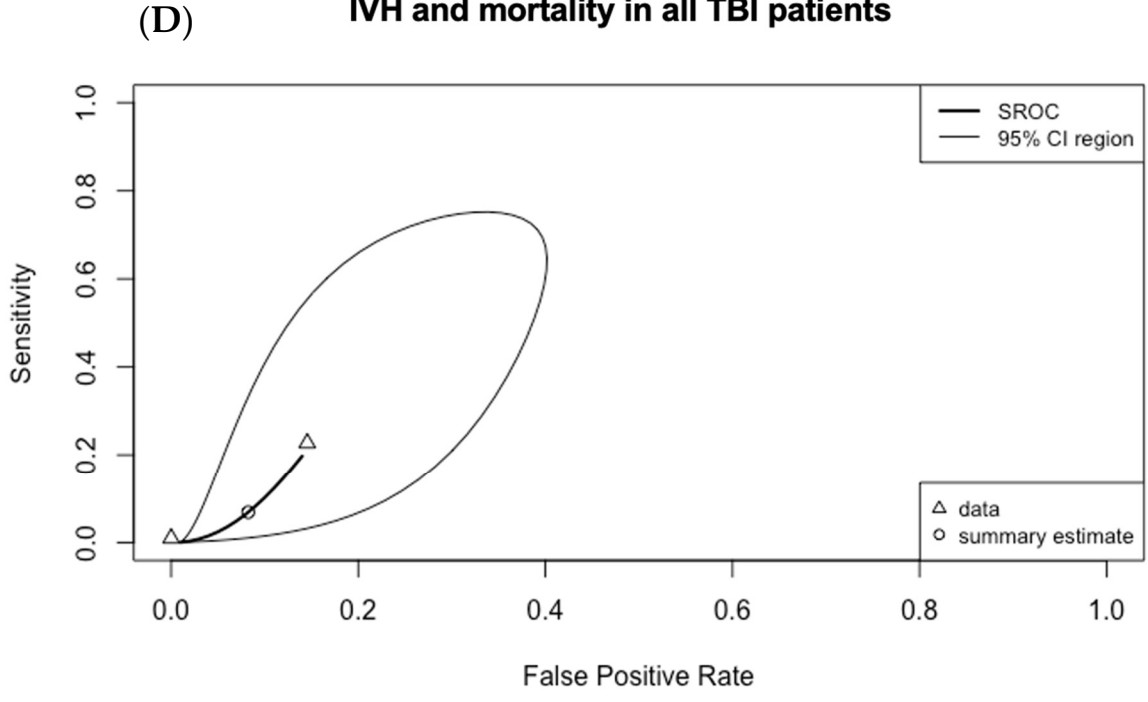

AUC: 0.722 (95% CI 0.051–0.925).

DOR: 1.461 (95% CI: 0.472–4.520).

Higgins' I²: 0%

**Figure 4.** *Cont.*

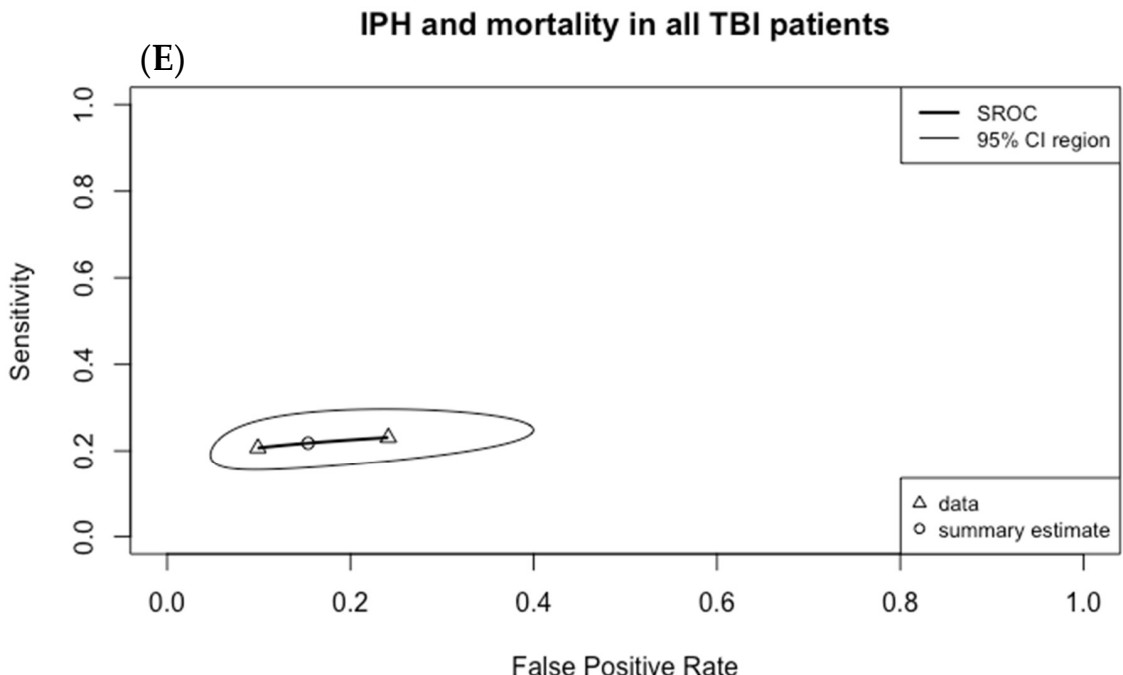

AUC: 0.259 (95% CI: 0.217–0.860).
DOR: 1.928 (95% CI: 0.909–4.088).
Higgins' I²: 0%

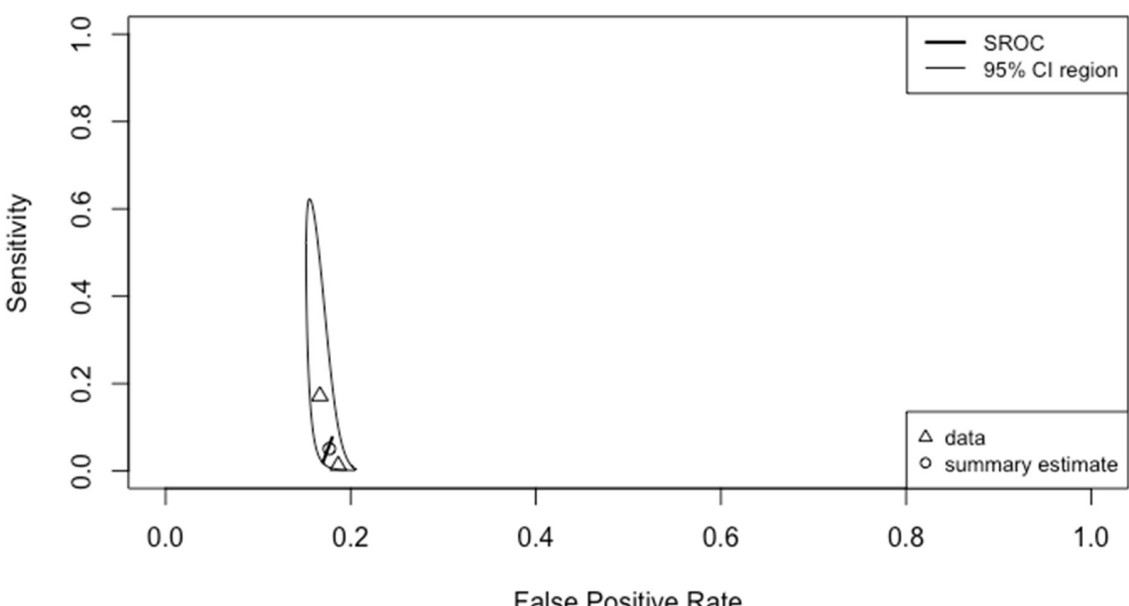

AUC: 0.799 (95% CI: 0.032–0.851).
DOR: 0.225 (95% CI: 0.013–4.010).
Higgins' I²: 0%

**Figure 4.** *Cont.*

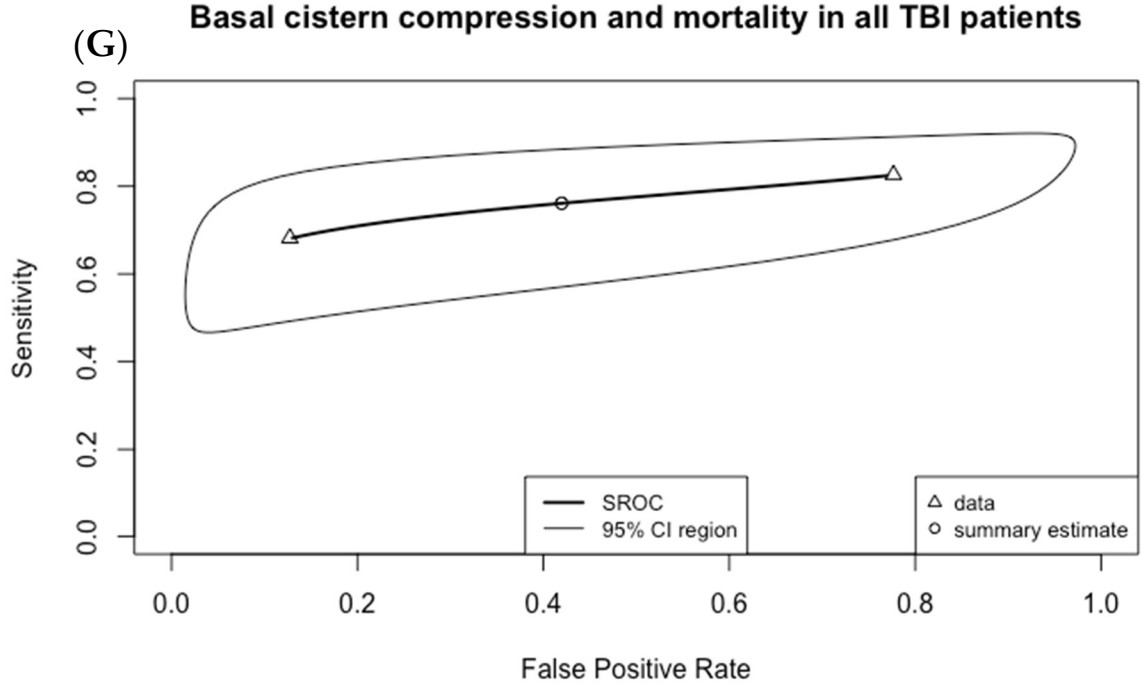

AUC: 0.760 (95% CI: 0.383–0.910).
DOR: 4.472 (95% CI: 0.436–45.858).
Higgins' I²: 0%

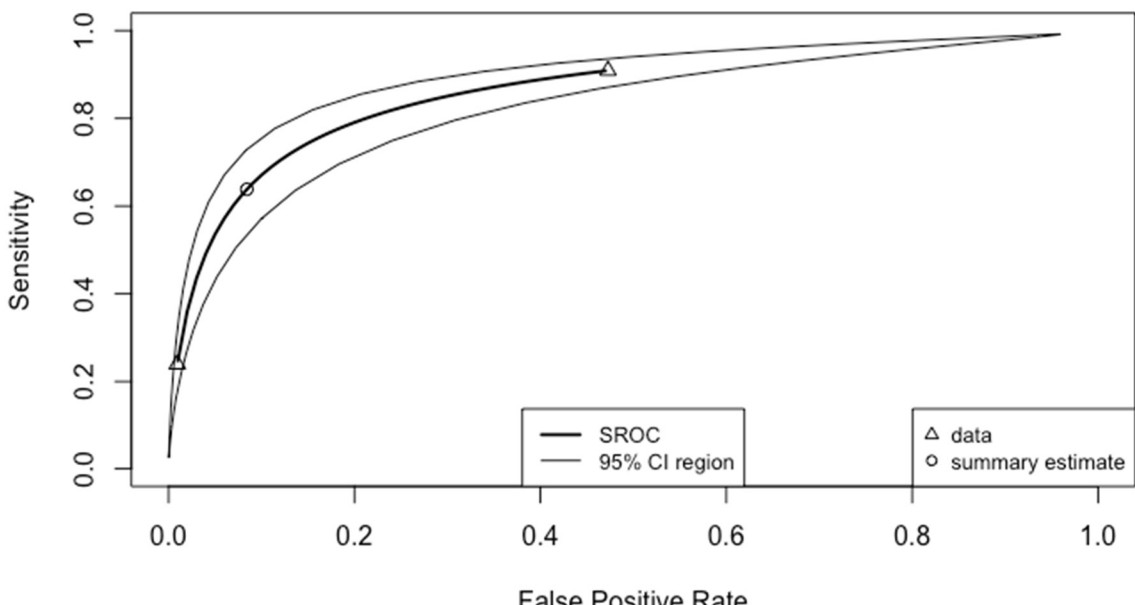

AUC: 0.860 (95% CI: 0.429–0.977).
DOR: 25.124 (95% CI: 9.986–63.211).
Higgins' I²: 0%

**Figure 4.** *Cont.*

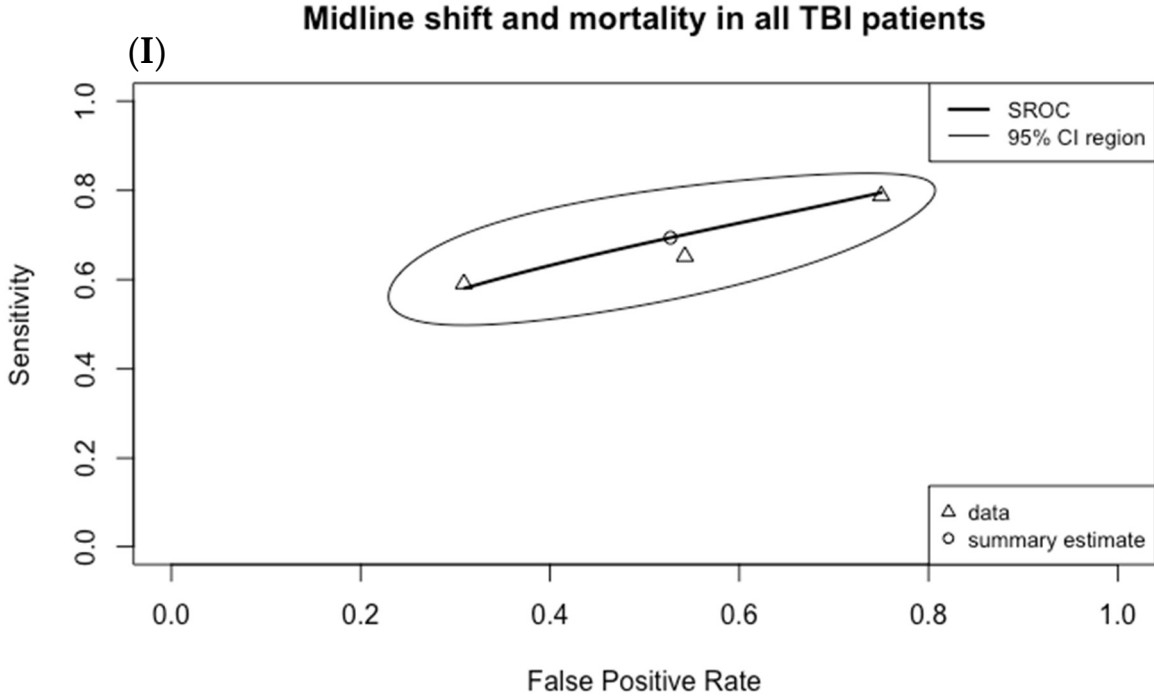

AUC: 0.650 (95% CI: 0.429–0.758).
DOR:1.960 (95% CI: 1.052–3.654).
Higgins' I²: 0%

**Figure 4.** Summary ROC curves (SROC; bivariate model) for mortality in adult TBI patients for (**A**) subarachnoid hemorrhage (SAH), (**B**) subdural hematoma (SDH), (**C**) epidural hematoma (EDH), (**D**) intraventricular hemorrhage (IVH), (**E**) intraparenchymal hematoma (IPH), (**F**) cerebral contusion, (**G**) basal cistern compression, (**H**) cerebral edema, and (**I**) midline shift. Only SDH had both an AUC with 95% CI not crossing 0.5 and a DOR with 95% CI not crossing 1, at 0.593 (95% CI: 0.556–0.725) and 2.755 (95% CI: 1.474–5.148), respectively.

## 4. Discussion

To our knowledge, this is the first systematic review and meta-analysis of initial imaging findings undertaken within 24 h post-presentation and their ability to prognosticate mortality in TBI patients. With NECT, the findings of our systematic review are generally in keeping with prior observations that a midline shift greater than 5 mm, effacement of the basal cisterns, and SAH are associated with increased mortality [6]. The findings of decreased MCA velocity and PI are also concordant with a prior systemic review and meta-analysis on TCD and outcomes in TBI performed by Fatima et al. 2019 [12].

For the pooled data, only SDH with mortality in adult TBI patients had a moderate but significant association with AUC of 0.593 (95% CI: 0.556–0.725) and DOR of 2.755 (95% CI: 1.474–5.148). No other imaging finding had both a statistically significant AUC and DOR. Although a sensitivity analysis was not pursued, this result may be secondary to the inclusion and exclusion criteria of requiring all patients to be 18 years of age or older, and for imaging to be done within 24 h of presentation to ED. The result can also be secondary to selection bias in the pooled data and to significant heterogeneity in the included studies, particularly regarding outcome measures and the patient population studied.

Our study focused solely on imaging characteristics without consideration of clinical findings such as decreased GCS and absence of pupillary reflex. Despite this, we believe that imaging features by themselves are valuable in prognostication as they are the only direct way to assess underlying anatomy and physiology and because of their wider availability, improved techniques, and faster scan times. Although prior multivariate

models combining clinical, biochemical, and imaging findings have been created, no model is validated for routine clinical use and prior systematic reviews and meta-analyses evaluating various multivariate prognostic models have concluded that the majority have poor external validation and quality [38–40].

There were several limitations to our review. All the included studies were observational, most provided experiences at a single centre, and there was significant heterogeneity in the studied imaging findings and outcome measures. There were a limited number of articles included that studied modalities that were not NECT, including only three studies regarding CTP and two involving TCD. Additionally, no studies investigated MRI and associated findings. MRI is a widely used modality for assessing the extent of traumatic brain injury and is significantly more sensitive than CT in the detection of cerebral injuries such as hemorrhagic cortical contusions and white matter shearing lesions, as well as in evaluating the temporal course of intracranial hemorrhage [5]. Similarly, there were no studies that used nuclear medicine examinations such as Tc-99m HMPAO cerebral perfusion SPECT [11]. The reasoning is favored secondary to their more limited availability and longer examination times; this means it would have been logistically difficult and unfeasible for TBI patients to have these examinations within 24 h of presentation as the care team would have been focused on resuscitation and stabilization. Furthermore, MRI is generally considered superior to CT at least 48–72 h post-TBI and in the subacute, chronic, and remote phases of injury, making scanning less than 24 h post-presentation even less likely [5,41]. Shankar et al. 2020 demonstrated high sensitivity and specificity, as well as high negative and positive predictive value in CTP's ability to prognosticate in-hospital mortality [30]. However, it was the only study of its kind included in our study and its data could not be used for meta-analysis. Finally, our study did not assess for improvements in scan speed, increased resolution, and improved techniques that could have aided in more accurate imaging diagnosis.

## 5. Conclusions

In conclusion, initial imaging findings of cerebral edema, midline shift, and SAH, the presence of decreased CBF and CBV, decreased MCA velocity and increased PI, and increased ONSD are all associated with mortality and unfavorable outcomes in TBI patients based on the available literature. In meta-analysis, only SDH with mortality in adult TBI patients had a moderate but significant association with AUC of 0.593 (95% CI: 0.556–0.725) and DOR of 2.755 (95% CI: 1.474–5.148). Given the small number of studies, additional research focused on initial imaging, particularly for imaging modalities other than NECT, is required in order to confirm the findings of our meta-analysis and to further evaluate the association between imaging findings and outcomes.

**Author Contributions:** Conceptualization, J.S., D.B., S.R.A. and H.Y.; methodology J.S., J.L. and H.Y.; software, H.Y.; validation, J.S., D.B., S.R.A. and H.Y.; formal analysis, J.S. and H.Y.; data curation, J.S. and H.Y.; writing—original draft preparation, J.S. and H.Y.; writing—review and editing, J.S. and H.Y.; visualization, H.Y.; supervision, J.S.; project administration, J.S. and J.L. All authors have read and agreed to the published version of the manuscript.

**Funding:** This research received no external funding.

**Institutional Review Board Statement:** Ethical review and approval were waived for this study as it uses publicly available documents and involves minimal risk.

**Informed Consent Statement:** Not applicable.

**Data Availability Statement:** The data presented in this study are available on request from the corresponding author.

**Conflicts of Interest:** The authors declare no conflict of interest.

## Abbreviations

| | |
|---|---|
| ACA | anterior cerebral artery |
| ARDS | acute respiratory distress syndrome |
| AUC | area under (the receiver operating characteristic) curve |
| CBF | cerebral blood flow |
| CBV | cerebral blood volume |
| CI | confidence interval |
| CT | computed tomography |
| CTA | CT angiography |
| CTP | CT perfusion |
| ED | emergency department |
| EDH | epidural hematoma |
| FU | follow-up |
| GCS | Glasgow Coma Score |
| GOS | Glasgow Outcome Scale |
| ICH | intracranial hemorrhage |
| ICP | intracranial pressure |
| ICU | intensive care unit |
| IPH | intraparenchymal hematoma |
| ISS | injury severity score |
| IVH | intraventricular hemorrhage |
| LR | likelihood ratio |
| MCA | middle cerebral artery |
| MRI | magnetic resonance imaging |
| MTT | mean transit time |
| NECT | non-enhanced CT |
| NPV | negative predictive value |
| ONSD | optic nerve sheath diameter |
| PET | positron emission tomography |
| PCA | posterior cerebral artery |
| PI | pulsatility index |
| PPV | positive predictive value |
| ROC | receiver operating characteristic |
| SAH | subarachnoid hemorrhage |
| SDH | subdural hematoma |
| SPECT | single-photon emission computed tomography |
| SROC | summary receiver operating characteristic |
| TBI | traumatic brain injury |

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
