# Peer review of "Prognostic Value of Initial Diagnostic Imaging Findings for Patient Outcomes in Adult Patients with Traumatic Brain Injury: A Systematic Review and Meta-Analysis"

_tomography, doi:10.3390/tomography9020042_

Round 1

Reviewer 1 Report

The review by Yu et al. specifically evaluated the prognostic value 12 of features on initial imaging and found only SDH with mortality in all TBI patients had 27 significant association. I think the quality of this manuscript is suitable to publish on Tomography after several corrections are made.

1, There are too many abbreviations in the abstracts, such as SDH, NECT etc.. it’s hard for readers to understand from the beginning, please define the abbreviations before you use it.

2, Line 27: Only SDH with mortality in all TBI patients had 27 significant association. Please quantify it in the abstract, how significantly association? What’s the level.

Author Response

We have made the following changes to the manuscript to address your concerns:

  • Made sure all abbreviations were relevant and defined within the manuscript.
  • Defined the association of SDH and mortality as a moderate association and provided the actual values for AUC and DOR in the abstract 

Reviewer 2 Report

The manuscript merits appreciation because it is the first systematic review and meta-analysis of early initial imaging findings in traumatic brain injury (TBI) patients and their ability to prognosticate mortality in TBI patients. Previous studies on this relevant problem already showed - having used ultrasound and CT, that edema, midline shift and subarachnoid hemorrhage are strongly correlated with bad outcome and mortality. The year of publication of the reviewed articles ranges from 2004 to 2017. There was no similar review published recently. The meta-analysis was made on 19 articles comprising more than 10000 patients.

The decision on the selection of the publications examined is clearly presented.

For pooled data in meta-analysis, forest plots for sensitivity and specificity were created to calculate the diagnostic odds ratio. The figures are apprpriate and they properly show the data, they are easy to understand and interpret. The language is excellent. The result of the investigation is surprising: Only sudural hematoma is correlated to mortality in a statistically significant manner.

The authors are not able to explain this result. The restriction to adult patients and to CT-imaging 24 hours after admission is not sufficient to explain this result. As this result is so astonishing and unique, it motivates to address further studies to try to corroborate or to correct the main finding. To do this, the article must be published. Hopefully, multicenter studies with MRI-imaging will follow and specify the prognosis of outcome by imaging in brain trauma patients.

Author Response

We have made the following changes to the manuscript to address your concerns:

  • Made sure all references cited were relevant
  • Thanks for your comments and we agree that this topic needs further research and publication of the current manuscript is key to future research.